# Study on Mechanical Properties of Hybrid Polypropylene-Steel Fiber RPC and Computational Method of Fiber Content

**DOI:** 10.3390/ma13102243

**Published:** 2020-05-13

**Authors:** Chunling Zhong, Mo Liu, Yunlong Zhang, Jing Wang, Dong Liang, Luyao Chang

**Affiliations:** 1School of Economics and Management, Jilin Jianzhu University, Changchun 130000, China; zhongchunling@jlju.edu.cn (C.Z.); 15237839994@163.com (L.C.); 2School of Civil Engineering, Jilin Jianzhu University, Changchun 130000, China; lium547573@163.com (M.L.); liangdong_d@163.com (D.L.); 3School of Transportation Science and Engineering, Jilin Jianzhu University, Changchun 130000, China

**Keywords:** reactive powder concrete, polypropylene fiber, steel fiber, compressive strength, splitting tensile strength

## Abstract

On the basis of determining the optimum content of polypropylene fiber reactive powder concrete (RPC), the influence of different steel fiber content on the compressive strength and splitting tensile strength of hybrid polypropylene-steel fiber RPC was studied. The particle morphology and pore parameters of hybrid polypropylene-steel fiber RPC were analyzed by combining scanning electron microscope (SEM) with image-pro plus (IPP) software. The results showed that the RPC ductility can be further improved on the basis of polypropylene fiber RPC, the compressive strength and splitting tensile strength of polypropylene fiber. The optimum content of hybrid polypropylene-steel fiber RPC is 0.15% polypropylene fiber, 1.75% steel fiber. Hybrid polypropylene-steel fiber RPC is mainly composed of particles with small particle size. The particle area ratio first increased and decreased with the increase of steel fiber content, and the maximum steel fiber content is 1.75%. The pore area ratio first decreased and increased with the increase of steel fiber content, and the pore area ratio is the smallest when the steel fiber content is 1.75%. The calculation methods of polypropylene fiber content and steel fiber content and 28-day RPC compressive strength and splitting tensile strength are proposed to select polypropylene fiber content and steel fiber content flexibly according to different engineering requirements, which can provide important guidance for the popularization and application of RPC in practical engineering.

## 1. Introduction

Reactive powder concrete (RPC) is a new type of building material with high strength and toughness developed by Richard in the 1990s. It removes coarse aggregate on the basis of ordinary concrete, and optimizes the compactness of aggregate with quartz sand, improving the homogeneity of matrix, and reducing porosity by adding active mineral admixtures such as silica fume and fly ash, while improving the microstructure by thermal curing [1,2,3]. In order to achieve ultrahigh strength of RPC, current research uses quartz sand [4,5,6] to prepare RPC through heat curing [7] or autoclave curing [8,9]. Thermal and autoclave curing can effectively improve RPC mechanical properties and promote its hydration process of RPC and enhance pozzolanic activity [10,11]. However, the cost of preparing RPC is increased because quartz sand cost is more expensive. The construction appears to be more difficult because of field thermal curing. These problems have limited the application of RPC and in order to expand its application, low cost common materials and conventional curing methods are adopted at the expense of certain reduction of strength.

Polypropylene fiber is a kind of non-toxic harmless safe material, which is cheap and has relatively low mass. Adding polypropylene fiber into RPC not only could improve strength and toughness, but also its crack resistance [12,13]. A lot of research suggests that polypropylene fiber RPC has limited effect on the compressive strength [14,15], while significantly improving the ductility of the RPC and preventing explosive spalling [16,17]. With the content from 0.3% to 0.9%, the addition of polypropylene fiber can improve the mechanical properties of RPC. With the dosage of more than 0.9%, the mechanical properties of RPC decreased to a certain extent [18]. Incorporation of steel fibers can significantly improve the mechanical properties of the RPC and change its failure morphology, and the ductility is significantly improved [19]. It is widely used in the preparation of RPC, the addition of steel fiber can reduce cracks and inhibit its broadening. However, the content of steel fiber should be paid attention. If the amount of steel fiber is too high, it will cause the steel fiber to occupy part of the mixing water, and the steel fiber lacks enough slurry to wrap and fill, making the steel fiber unevenly distributed and agglomerated, which in turn leads to the steel fiber’s contribution to improving strength not increasing, but decreasing [20]. Due to the different curing methods of RPC and the different sizes of steel fibers, the obtained optimal amount of steel fiber is also different. When the amount of steel fiber is increased from 0% to 3%, the compressive strength of RPC is slightly increased [21]. However, after the amount of steel fiber is more than 3.5%, the compressive strength of RPC does not increase any more. The splitting tensile strength increases obviously when the steel fiber is less than 2%, and the splitting tensile strength remains almost unchanged over 2% [22]. The results showed that the splitting tensile strength increased from 4.033 to 9.609, 11.944, and 12.896 MPa, respectively when the steel fiber increased from 0% to 1%, 1.5%, and 2% [23]. The results showed that with the content of steel fiber from 0% to 4%, the compressive strength and splitting tensile strength are increased, and the compressive strength of steel fiber to RPC is increased less, and the splitting tensile strength is obviously improved [20,24]. Moreover, the effect of incorporation of different types of steel fibers on the mechanical properties of the RPC is also different. The compressive strength can reach 150 MPa or more when 3% ordinary steel fiber is blended, and the 28-day compressive strength of RPC with 3% end hook and corrugated fiber is increased by 48% and 59%, respectively [25]. Hence, it is necessary to further study the mechanical properties of polypropylene-steel fiber RPC.

Concrete is a complex multi-phase heterogeneous material, and factors such as temperature and stress will induce phase transformation [26,27]. It is difficult to make reliable analysis of concrete only through macroscopic mechanical properties. Therefore, it is necessary to analyze the macroscopic properties of concrete through microscopic research to determine the relationship between the microstructure and macroscopic properties of concrete, which is of great significance to the research and development of new concrete materials. Studies have shown that the particle morphology and pore parameters of aggregate have a significant effect on the compressive strength of concrete [28,29]. Some scholars use XCT combined phase field theory to analyze the microscopic parameters of materials, but the computational cost involved is high [30]. The IPP (Image-pro plus) software has powerful image processing function, which is not only simple to operate but also low cost. Therefore, the microscopic parameters of polypropylene-steel fiber RPC are analyzed by combining scanning electron microscope (SEM) with image-pro plus (IPP) software (SEM-IPP).

In summary, previous studies mainly focus on the RPC mechanical properties of single-doped polypropylene fiber and single-doped steel fiber, and the mechanical properties of hybrid polypropylene-steel fiber RPC are less studied. Since the hybrid fibers have certain complementarity and synergy, the high–modulus steel fibers can effectively enhance the mechanical properties of RPC, while the low-modulus polypropylene fibers have certain advantages in improving the toughness of RPC. On the basis of determining the optimum content of polypropylene fiber RPC, the influence of different steel fiber content on the mechanical properties of hybrid polypropylene-steel fiber was studied. By combining SEM with IPP software, the particle and pore parameters of hybrid polypropylene-steel fiber RPC were analyzed. On this basis, the computing method of polypropylene fiber content, steel fiber content and compressive strength and the computing method of polypropylene fiber content, steel fiber content and splitting tensile strength are presented and can be used to provide a reliable polypropylene fiber content and steel fiber content scheme, which can provide reference for the preparation of economical and reasonable RPC in practical engineering.

## 2. Materials and Methods

### 2.1. Raw Materials

This study adopted P.II-grade 52.5 cement. The cement inspection report, which conforms to the general Portland Cement Inspection Standard (GB175-2007) [31], is shown in Table 1. Silica fume and fly ash were used. The test reports of silica fume are shown in Table 2. The test reports of fly ash are shown in Table 3. All these materials were in accordance with the technical specifications of the Application of Mineral Admixtures (GB/T51003/2014) [32]. The superplasticizer used was HSC polycarboxylic acid high-performance water reducer. The inspection indexes are shown in Table 4. The fine aggregate was made of river sand with particle size below 1.18 (Changchun, China). The screening results, shown in Table 5, are in accordance with the Standard for Technical Requirements and Test Method of Sand and Crushed Stone for Ordinary Concrete (JGJ 52-2006) [33]. The gradation curve is shown in Figure 1. The polypropylene fiber used in this work was produced by Hongju Engineering Materials Co., Ltd. (Taian, China). The steel fiber was copper-plated steel fiber produced by Zhitai Steel Fiber Industry (Tangshan, China). The specifications and performance indexes of the polypropylene fiber and steel fiber are shown in Table 6, and corresponding images are shown in Figure 2.

### 2.2. Sample Preparation

The mixing of RPC differs from the conventional concrete mixing process. Given the low water-binder ratio of RPC itself, the addition of fiber increases its mixing difficulty. To ensure the uniform distribution of polypropylene fiber and steel fiber and reduce the impact of fiber agglomeration on RPC performance, the mixture was added in sections according to the different types of fiber. A standard curing box with temperature of 20 °C and humidity of 95% was used in the test. After 24 h of maintenance, the mold was removed and continued to be maintained under standard curing conditions for 7 and 28 days.

### 2.3. Testing Procedure

#### 2.3.1. Mechanical Properties Tests

Specimens (100 × 100 × 100 mm) were selected as test pieces on the basis of GB/T31387-2015 [34] to evaluate the compressive and splitting tensile strengths. Before the test, wipe the specimen surface and the press upper and lower pressure plate surface clean and place on the press plate to align the specimen center with the press plate center. The samples were tested with SYE-3000B (hydraulic press of New Testing Machine Co., Ltd. (Changchun, China) for 7 and 28 days. The average of three measurements was used as the compressive strength and splitting tensile strength of the test specimens. The compressive test loading rate was maintained between 1.2 and 1.4 MPa/s, and the loading rate of the splitting tensile strength test was maintained between 0.08 and 0.1 MPa/s.

#### 2.3.2. Microstructural Tests

Samples for SEM were small pieces (about 5 mm) taken from the compressive failure specimens. They were then polished with sandpaper, and finally with a polished glass plate and a polishing cloth. Through vacuum drying and gold spraying, the hybrid polypropylene-steel fiber RPC microstructures were observed and photographed using the environmental scanning electron microscope of FEI Company (Hillsboro, OR, USA). Vacuum drying and gold spraying are shown in Figure 3.

#### 2.3.3. Digital Image Processing

The image processing function of the Matlab analysis software was used to process the hybrid polypropylene-steel fibers RPC microscopic images, including two parts: image enhancement and image segmentation.

Image enhancement: Histogram equalization is to make the image clearer and brighter by adjusting the contrast of the image. The method of using median filter to remove the image in the process of imaging, transmission interference, as shown in Figure 4, after equalization and filtering denoising images, while improving the overall quality of the image, but the image of the local position is still the ambiguous part, using the image sharpening to supplement in the position of the local fuzzy, change the contrast, to strengthen the image, make it clear, as shown in Figure 5.Image segmentation: In order to be able to hybrid polypropylene-steel fibers RPC target region and background region division, using the method of threshold segmentation in the region of interest in the image according to the gray level range segmentation, as shown in Figure 6.

## 3. Results and Discussion

### 3.1. Experimental Results and Analysis of Mechanical Properties of Polypropylene Fiber RPC

#### 3.1.1. Experiment Design

Polypropylene fibers with volume contents of 0%, 0.15%, 0.3%, 0.45%, and 0.6% were added on the basis of the matrix mix proportions of RPC. Polypropylene fiber RPC mix proportions are shown in Table 7. The mechanical properties of the polypropylene fiber RPC with different polypropylene fiber contents were studied.

#### 3.1.2. Polypropylene Fiber RPC Failure Mode

The compression failure pattern of polypropylene fiber RPC is shown in Figure 7. As observed in the compression test, it did not show the mixing of the polypropylene fibers of RPC in the early stages of the load. Numerous runs slowly emerged through the whole crack on the surface of each specimen as the load continued to increase. Extensive peeling was also observed, along with damage due to noise. Bare internal RPC substrates were noted as well. The pattern belonged to the brittle failure mode. When the polypropylene fiber was added, the polypropylene fiber constrained the test block. As the load increased, the RPC began to expand, and only a small number of fragments fell off, resulting in different degrees of cracks. Perforation cracks were ultimately formed.

The failure pattern of the splitting tensile strength of polypropylene fiber RPC is shown in Figure 8. In the absence of polypropylene fibers in the RPC specimens, vertical fracturing was observed. With the addition of polypropylene fiber, the specimens no longer presented brittle fracturing but showed a flat interface crack. Crack initiation and carry out the first appeared in the tensile strength of the weak base inside, in the event of micro cracks, matrix together with polypropylene fiber under tensile stress. As stress increased, the microcrack width increased gradually. Finally, the tensile stress is transferred to the concrete on both sides of the crack by the adhesive stress of polypropylene fiber and matrix. When the tensile stress exceeded the bonding stress between the polypropylene fiber and the substrate, bonding failure occurred.

#### 3.1.3. Test Results of Mechanical Properties of Polypropylene Fiber RPC

The test results of compressive strength and splitting tensile strength of different contents of polypropylene fibers for 7 and 28 days for are shown in Table 8.

#### 3.1.4. Analysis of the Mechanical Properties of Polypropylene Fiber RPC

The test results for the compressive and splitting tensile strengths of polypropylene fiber RPC are shown in Figure 9 and Figure 10, respectively.

Figure 9 and Figure 10 indicate that with the increase of polypropylene fiber content, the compressive strengths and splitting tensile strengths of polypropylene fiber RPC for 7-day and 28-day first increased and decreased. The 7-day maximum compressive strength of RPC was 0.15% polypropylene fiber. The 7-day maximum splitting tensile strength of RPC was 0.3% polypropylene fiber. The 28-day highest compressive and splitting tensile strengths of polypropylene fiber RPC were 0.15% polypropylene fiber, the maximum compressive strength was 96.12 MPa, and the maximum splitting tensile strength was 12.89 MPa. Therefore, the optimal content of polypropylene fiber of RPC was 0.15%.

### 3.2. Experimental Results and Analysis of Mechanical Properties of Hybrid Polypropylene-Steel Fiber RPC

#### 3.2.1. Experiment Design

On the basis of the optimal content of polypropylene fiber RPC, 0%, 1%, 1.25%, 1.5%, 1.75%, and 2% of steel fiber were added. Hybrid polypropylene-steel fiber RPC mix proportions are shown in Table 9. The mechanical properties of hybrid polypropylene-steel fiber RPC were further studied.

#### 3.2.2. Hybrid Polypropylene-Steel Fiber RPC Failure Mode

The failure pattern of the compressive strength of hybrid polypropylene-steel fiber RPC is shown in Figure 11. In the process of compression, the surface of the specimen does not appear through cracks, but a large number of fine cracks that are not through, indicating that the characteristics of high toughness of steel fiber inhibit the extension of cracks. After failure, the RPC specimens basically maintained their original integrity, and only many cracks and peeling appeared, which further improved the plastic properties of the RPC. The failure pattern of the splitting tensile strength of hybrid polypropylene-steel fiber RPC is shown in Figure 12. The influence mechanism of the split tensile failure of mixed polypropylene and steel fiber RPC lies in the bridging action of steel fiber and polypropylene fiber. In the splitting tensile failure test, the fracture section is basically a flat straight line, and no penetrating fracture is formed. The specimen remains as a whole after the failure. The microcracks of mixed hybrid polypropylene-steel fiber RPC first appeared in the weak matrix. As the stress increases, the load is borne by the bonding force between the fiber and the cement slurry. When the stress exceeds the bonding stress between the fiber and the substrate, the bond failure occurs.

#### 3.2.3. Test Results of Mechanical Properties of Hybrid Polypropylene-Steel Fiber RPC

The test results of compressive and splitting tensile strengths of hybrid polypropylene-steel fiber RPC under different steel fiber contents for 7 and 28 days are shown in Table 10.

#### 3.2.4. Analysis of Mechanical Properties of Hybrid Polypropylene-Steel Fiber RPC

Figure 13 and Figure 14 show the resulting compressive strengths and splitting tensile strengths of hybrid polypropylene-steel fiber RPC for 7 and 28 days, respectively.

Figure 13 and Figure 14 shows that with the increase of steel fiber content, the 7- and 28-day compressive and splitting tensile strengths of hybrid polypropylene-steel fiber RPC first increased and decreased. When the steel fiber content was 1.75%, the compressive and splitting tensile strengths were the highest. For hybrid polypropylene-steel fiber RPC, 7- and 28-day compressive strengths were 100.96 and 123.92 MPa, respectively, and the splitting tensile strengths were 16.19 and 25.19 MPa, respectively. When the steel fiber content was 2%, the compressive and splitting tensile strengths of the hybrid polypropylene-steel fiber RPC decreased mainly because of the increase in fiber volume content. The specific surface area of the steel fiber increased greatly and could not be adequately wrapped by slurry. This condition resulted in the uneven dispersion of the steel fiber in the hybrid polypropylene-steel fiber RPC. This uneven dispersion led to agglomeration, which in turn decreased the density of the RPC matrix and the compressive strength of the hybrid polypropylene-steel fiber RPC.

The percentage increase in compressive and splitting tensile strengths of hybrid polypropylene-steel fiber RPC with the increase in steel fiber content was analyzed by using undoped steel fiber as a reference, are shown in Figure 15 and Figure 16.

The effects of different steel fiber contents on the compressive and splitting tensile strengths of hybrid polypropylene-steel fiber RPC showed variations. The compressive strength of hybrid polypropylene-steel fiber RPC was improved by steel fiber in a small range. For hybrid polypropylene-steel fiber RPC7d and RPC28d were 1.75% steel fiber, the compressive strengths showed maximum percentage increases of 18.61% and 28.92%, respectively. As for their splitting tensile strengths, the values indicated large increases as the content of steel fiber increased. The percentage increase of the splitting tensile strength of hybrid polypropylene-steel fiber RPC7d increased gradually and peaked at 127.93%. The percentage increase of the splitting tensile strength of hybrid polypropylene-steel fiber RPC28d peaked at 97.75% when the steel fiber content was 1.75%. Steel fiber clearly improved the strength of hybrid polypropylene-steel fiber RPC because the strength of steel fiber is greater than that of the RPC matrix and its size is larger than the crack tip. Therefore, steel fiber can play the role of bridge and pin plug and hinder the development of cracks. Moreover, a moderate amount of steel fiber can increase the intensity of hybrid polypropylene-steel fiber RPC to a certain extent. However, if the steel fiber content is excessive, steel fiber clusters can form and reduce the strength of hybrid polypropylene-steel fiber RPC.

### 3.3. Microscopic Mechanism Analysis of Hybrid Polypropylene-Steel Fiber RPC

#### 3.3.1. Microimage Analysis of Hybrid Polypropylene-Steel Fiber RPC

The mechanical properties of hybrid polypropylene-steel fiber RPC are closely related to its microstructure. Therefore, the relationship between the mechanical properties and the microstructure of hybrid polypropylene-steel fiber RPC was further studied through the microscopic test of hybrid polypropylene-steel fiber RPC studied by SEM.

The fiber-matrix interface of hybrid polypropylene-steel fiber RPC is the weak link, as shown in Figure 17a. The structure of the fiber and matrix showed no obvious interface transition zone and exhibited good compactness, thereby improving the strength of the fiber matrix interface. However, when the steel fiber content was 2%, the steel fiber consumed a portion of the mixing water, resulting in the lack of slurry package that led to the formation of steel fiber clusters (Figure 17b). The steel fiber between the erection function also contributed to the increase of internal friction and reduced the strength of the mixture. Therefore, the compressive and splitting tensile strengths of RPC can be improved by mixing polypropylene fiber and steel fiber, but the content should be controlled within an appropriate range.

#### 3.3.2. Parameter Analysis of Hybrid Polypropylene-Steel Fiber RPC Particle

Fine aggregates are present in concrete and mineral admixture in cement condensation sclerosis. The surface of these aggregates is covered by a large amount of hydration products and slurry, the connection between each other forms the main body. Therefore, a quantitative analysis of polypropylene granules forming inside the steel fiber RPC should be carried out. In the present work, the influence law of strength on structural form was evaluated on the basis of quantitative stereology theory. The IPP image processing technology was used to investigate the microscopic particles of hybrid polypropylene-steel fiber RPC with different steel fiber contents. The parameters extracted are shown in Table 11 and Table 12.

The results of the microparticle parameters in Table 11 and Table 12 are mapped in Figure 18 and Figure 19, respectively.

For the hybrid polypropylene-steel fiber RPC with 0–2 μm particles, the small particles gradually expanded with the increase of steel fiber content. The hybrid polypropylene-steel fiber RPC with 2–5 and 5–20 μm particles showed no obvious regularity, but that with 0–5 μm particles showed an obvious expansion with the increase of steel fiber content. Particles greater than 5 μm gradually decreased. The particle area ratio gradually increased with the increase of the content of steel fiber in the range 0–1.75% and decreased when the content of steel fiber was 2%. The average particle size gradually increased with the increase of steel fiber content.

#### 3.3.3. Pore Parameter Analysis of Hybrid Polypropylene-Steel Fiber RPC

The microscopic properties of concrete, as a type of porous material, are closely related to its microstructure. Pore structure is an important part of the microstructure of cement concrete. Therefore, to understand the relationship between the macroscopic mechanical behavior and microstructure parameters of hybrid polypropylene-steel fiber RPC, this study extracted the pore microstructure parameters of hybrid polypropylene-steel fiber RPC with different steel fiber contents using IPP (Table 13 and Table 14). The results should reveal the relationship between the microstructure and strength of RPC.

The microparticle parameters in Table 13 and Table 14 are respectively drawn in Figure 20 and Figure 21.

According to the results, you can see that the porosity of the 0–4 μm is more, with the increase of steel fiber dosage, 0–4 μm of pore ratio first decreases and the trend of increase, steel fiber content was 1.75% of the minimum, and less than 4 μm pore, with the increase of steel fiber content, greater than 4 μm of pore ratio showed a trend of decrease after the first increase and steel fiber content with 1.75% proportion is the largest. The strength still improved even in the presence of abundant macropores. Specifically, steel fiber reduces the fluidity of mixing materials, thereby increasing macropore content. Nevertheless, steel fiber can improve material strength. The pore area ratio gradually decreased with the increase of the content of steel fiber within 0–1.75%, by contrast, it increased when the content of steel fiber was 2%. The average pore diameter gradually increased with the increase of the content of steel fiber.

### 3.4. Hybrid Polypropylene-Steel Fiber RPC Content Computational Methods

#### 3.4.1. Compressive Strength Computational Methods

In order to design the calculation method of the content of polypropylene fiber and the content of steel fiber and the compressive strength of hybrid polypropylene-steel fiber RPC, the contribution degree of polypropylene fiber was fitted by the percentage increase of the compressive strength of polypropylene fiber RPC at 28 days $\lambda_{cp}$ as shown in Figure 22. The percentage increase of compressive strength of hybrid polypropylene-steel fiber RPC at 28 days was used to fit the contribution of steel fiber $\lambda_{cs}$ as shown in Figure 23. The formula for calculating the content of polypropylene fiber, the content of steel fiber and the 28-day compressive strength of hybrid polypropylene-steel fiber RPC is put forward:(1)$$f_{c}=f_{0}(1+\lambda_{cp})(1+\lambda_{cs}),$$
where *f*_c_ is 28-day compressive strength of hybrid polypropylene-steel fiber (MPa), *f*_0_ corresponds to the 28-day compressive strength of the undoped fiber (MPa), $\lambda_{cp}$ is the polypropylene fiber compressive strength contribution (%), $\lambda_{cs}$ is the steel fiber compressive strength contribution (%).

The experimental results in this study were compared with the calculated results to verify the applicability of the hybrid polypropylene-steel fiber RPC compressive strength formula, as shown in Table 15.

Feng [35] investigated the mechanical properties of hybrid polypropylene-steel fiber RPC. To further verify the feasibility of the calculation method, the calculated value of this calculation method was compared with the test value [35], as shown in the Table 16.

According to Table 15 and Table 16, the error of the calculated value and actual value are not more than 15%, the maximum error is 13.72%, using the calculation method of calculation value and experimental results conform to the degree of good, therefore, we can use this method to calculate the predicting 28-day compressive strength hybrid polypropylene-steel fiber RPC, for RPC provides certain a guiding role in the practical engineering application.

#### 3.4.2. Splitting Tensile Strength Computational Methods

In order to design the calculation method of the content of polypropylene fiber and the content of steel fiber and the splitting tensile strength of hybrid polypropylene-steel fiber RPC, the contribution degree of polypropylene fiber was fitted by the percentage increase of the splitting tensile strength of polypropylene fiber RPC at 28 days $\lambda_{tp}$ as shown in Figure 24. The percentage increase of splitting tensile strength of hybrid polypropylene-steel fiber RPC at 28 days was used to fit the contribution of steel fiber $\lambda_{ts}$ as shown in Figure 25. The formula for calculating the content of polypropylene fiber, the content of steel fiber and the 28-day splitting tensile strength of hybrid polypropylene-steel fiber RPC is put forward:(2)$$f_{t}=f_{0}(1+\lambda_{tp})(1+\lambda_{ts}),$$
where *f_t_* is 28-day splitting tensile strength of hybrid polypropylene-steel fiber(MPa), *f*_0_ corresponds to the 28-day splitting tensile strength of the undoped fiber, $\lambda_{tp}$ is the polypropylene fiber splitting tensile strength contribution (%), $\lambda_{ts}$ is the steel fiber splitting tensile strength contribution (%).

In order to verify the hybrid polypropylene-steel fiber RPC splitting tensile strength of the applicability of the formula, as a result of the study, the splitting tensile strength of hybrid polypropylene-steel fiber RPC is less, therefore, we cannot use the other research conclusion, only using the experimental data validated, the calculated values and experimental values were analyzed, as shown in Table 17.

Table 17 shows that the error between the calculated value and the actual values are within 15%, the maximum error is 11.75%, compared with the experimental results conforming to the degree of good, it shows that this formula is suitable for polypropylene, the calculation of steel fiber RPC at 28-day splitting tensile strength, the strength required according to actual engineering, the choice of flexible of polypropylene fiber, and steel fiber content.

## 4. Conclusions

On the basis of the optimal content of polypropylene fiber in RPC, the mechanical properties of hybrid polypropylene-steel fiber RPC with different contents of steel fiber were studied. By combining SEM with IPP software, the particle and pore parameters of hybrid polypropylene-steel fiber RPC were analyzed along with the computing method of content of polypropylene fiber and steel fiber, as well as the 28 days compressive strength and splitting tensile strength of RPC. The following conclusions were drawn:Compared with single-doped polypropylene fiber RPC, hybrid polypropylene-steel fiber RPC has obvious improvement in compressive strength and splitting tensile strength, which can further improve the ductility of RPC on the basis of polypropylene fiber RPC.The optimum fiber content of hybrid polypropylene-steel fiber RPC is 0.15% polypropylene fiber and 1.75% steel fiber. The effect of mixed fiber on the increase of RPC compressive strength is small, while the splitting tensile strength is obviously improved.Hybrid polypropylene-steel fiber RPC is mainly composed of small particle size, the particle area ratio first increased and decreased with the increase of steel fiber content, and the maximum amount of steel fiber content is 1.75%. The pore area ratio first decreased and increased with the increase of steel fiber content. When the steel fiber content is 1.75%, the pore area ratio is the smallest.The formulae for calculating the content of polypropylene fiber and steel fiber in RPC, the compressive strength and splitting tensile strength are presented, which are in good agreement with the test results, and the content of polypropylene fiber and steel fiber can be flexibly selected according to the actual requirements of the project.In order to provide a reliable reference for the structural design of hybrid polypropylene-steel fiber RPC, in future work, the hybrid polypropylene-steel fiber RPC elastic modulus, deformability, and Poisson coefficient should be studied. The size effect has an important influence on the destruction process of concrete components [36]. It is necessary to further study the effect of the size effect of hybrid polypropylene-steel fiber RPC components, which has important practical significance for the structural design of hybrid polypropylene-steel fiber RPC.

## Figures and Tables

**Figure 1 materials-13-02243-f001:**
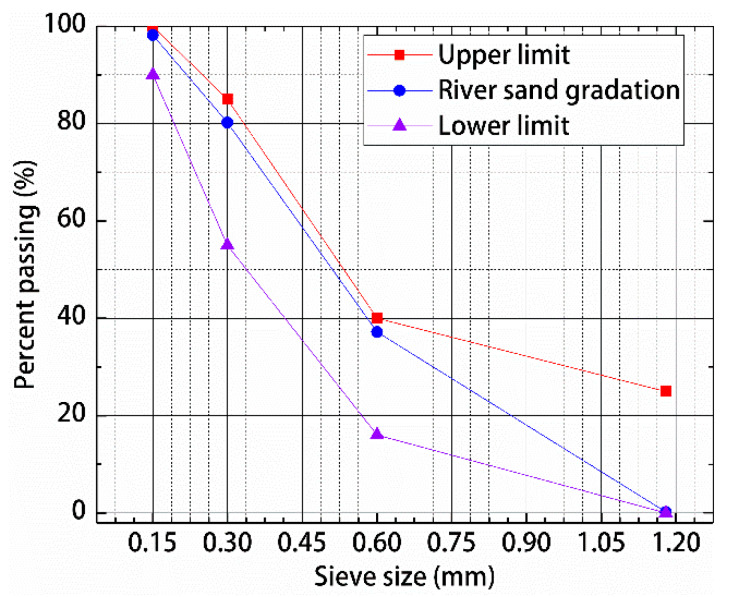
Gradation of river sand used in this study.

**Figure 2 materials-13-02243-f002:**
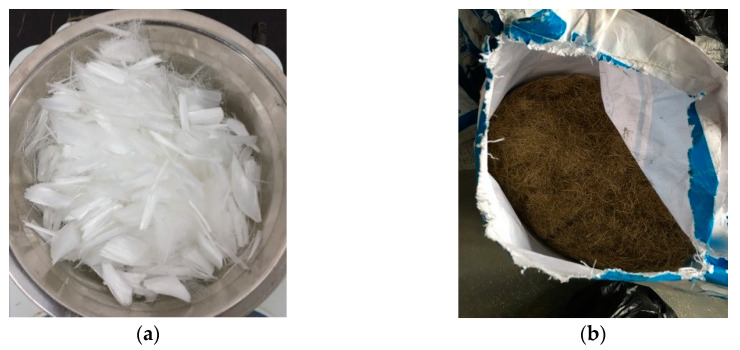
Fibers used in this study: (**a**) polypropylene fiber and (**b**) steel fiber.

**Figure 3 materials-13-02243-f003:**
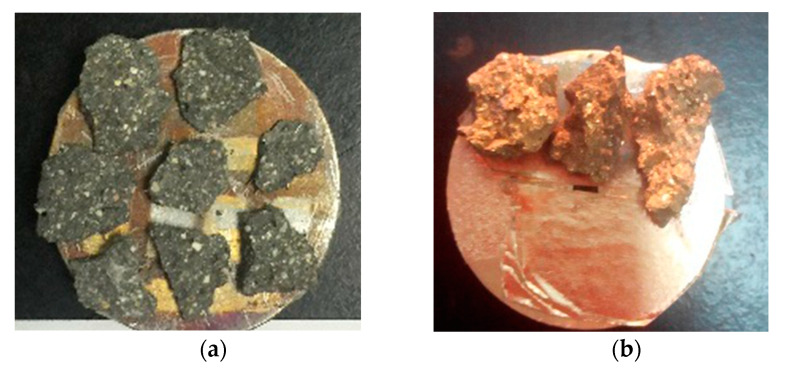
Specimen processing: (**a**) vacuum drying and (**b**) gold spraying.

**Figure 4 materials-13-02243-f004:**
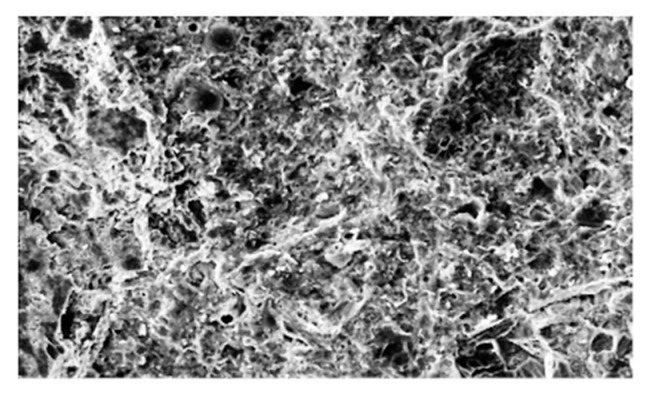
Image noise reduction.

**Figure 5 materials-13-02243-f005:**
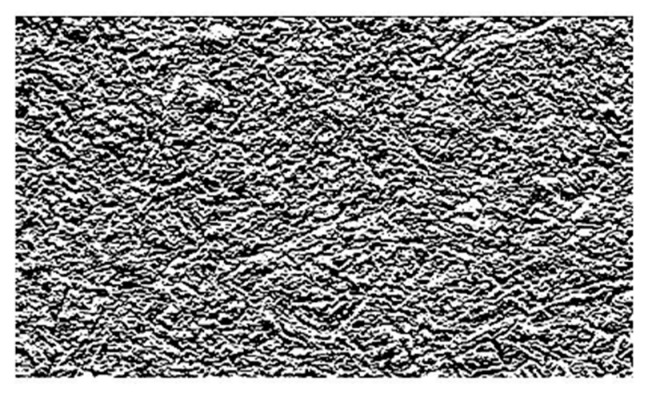
Image sharpening.

**Figure 6 materials-13-02243-f006:**
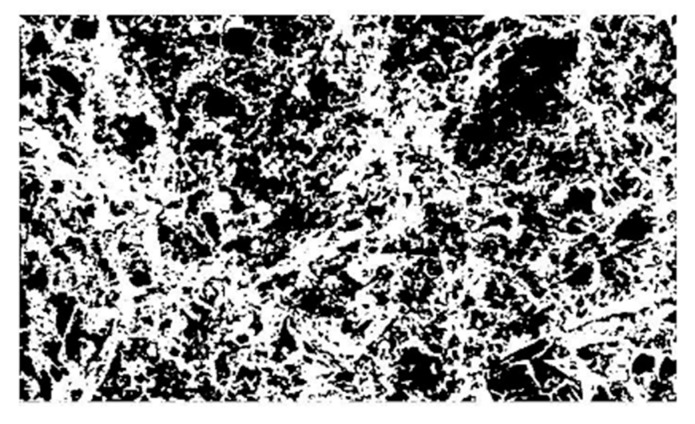
Image segmentation.

**Figure 7 materials-13-02243-f007:**
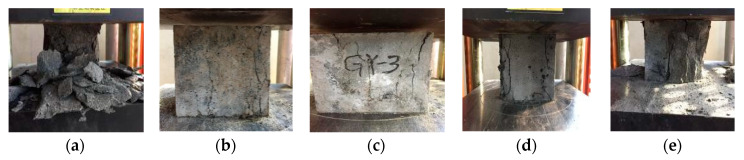
Compressive strength failure mode of polypropylene fiber RPC: (**a**) 0%, (**b**) 0.15%, (**c**) 0.3%, (**d**) 0.45%, and (**e**) 0.6%.

**Figure 8 materials-13-02243-f008:**
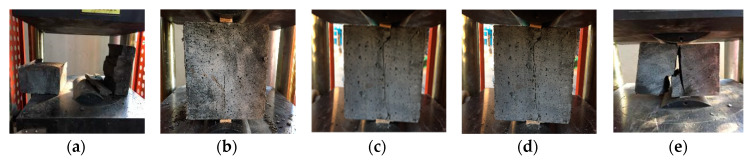
Splitting tensile strength failure mode of polypropylene fiber RPC: (**a**) 0%, (**b**) 0.15%, (**c**) 0.3%, (**d**) 0.45%, and (**e**) 0.6%.

**Figure 9 materials-13-02243-f009:**
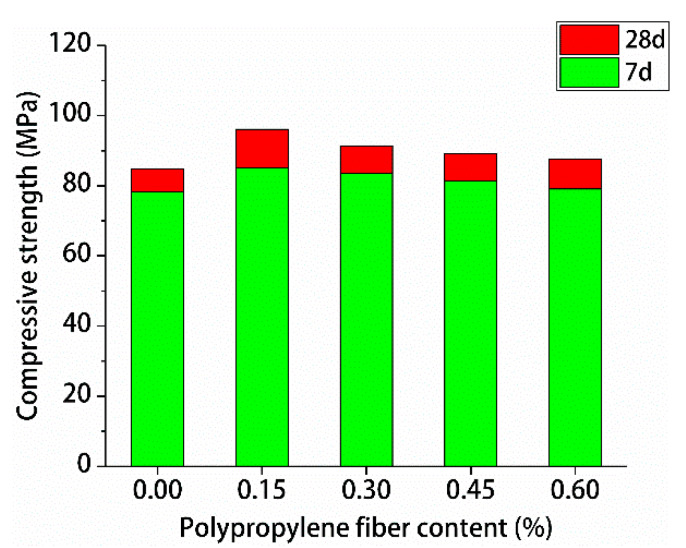
Test results for compressive strength.

**Figure 10 materials-13-02243-f010:**
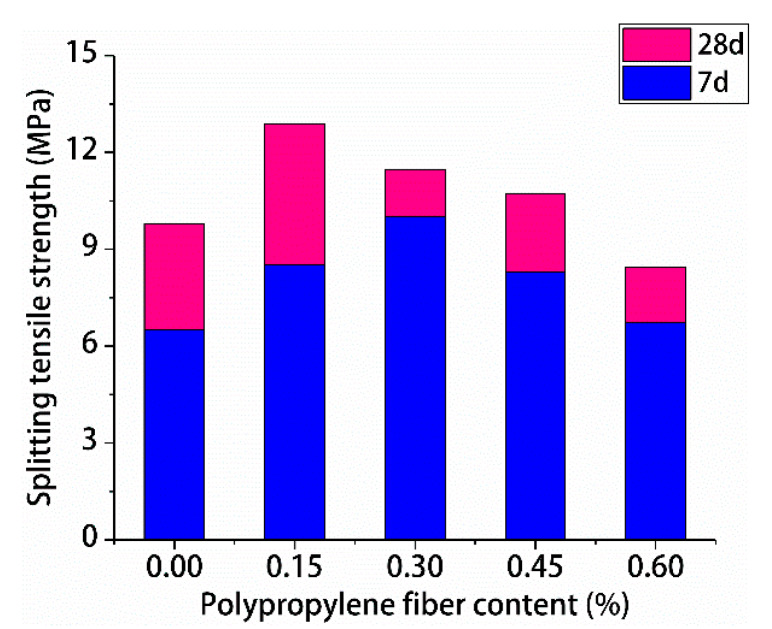
Test results for splitting tensile strength.

**Figure 11 materials-13-02243-f011:**
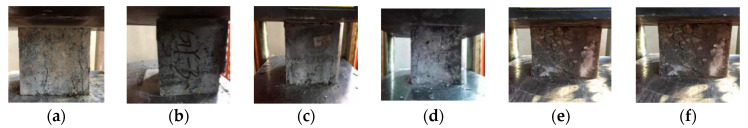
Compressive strength failure mode of hybrid polypropylene-steel fiber RPC: (**a**) 0%, (**b**) 1%, (**c**) 1.25%, (**d**) 1.5%, (**e**) 1.75 and (**f**) 2%.

**Figure 12 materials-13-02243-f012:**
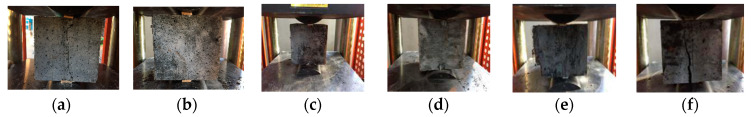
Splitting tensile strength failure mode of hybrid polypropylene-steel fiber RPC: (**a**) 0%, (**b**) 1%, (**c**) 1.25%, (**d**) 1.5%, (**e**) 1.75 and (**f**) 2%.

**Figure 13 materials-13-02243-f013:**
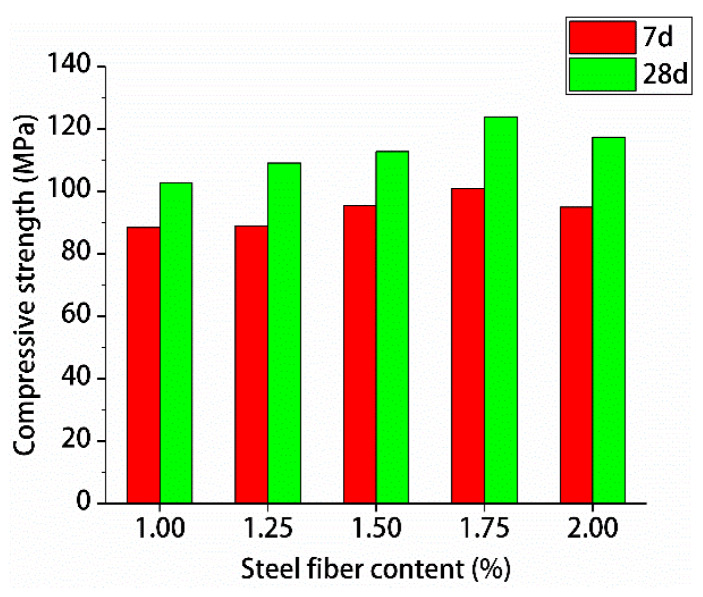
Test results for compressive strength.

**Figure 14 materials-13-02243-f014:**
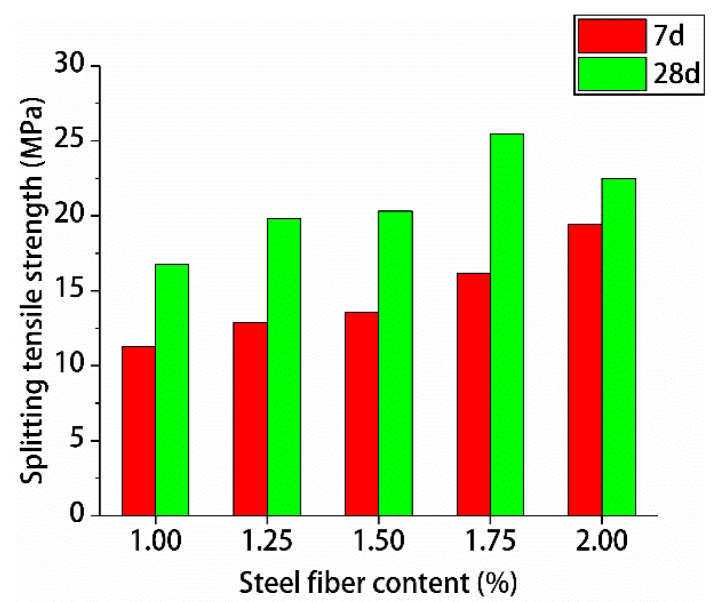
Test results for splitting tensile strength.

**Figure 15 materials-13-02243-f015:**
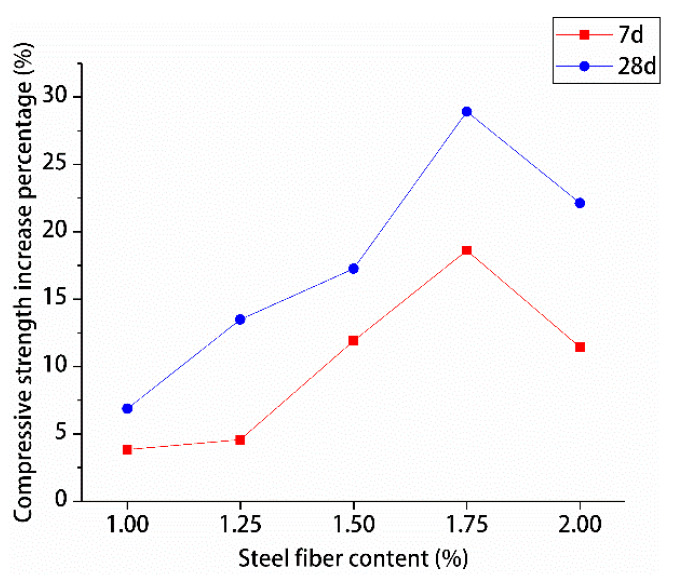
Increase percentage of compressive strength.

**Figure 16 materials-13-02243-f016:**
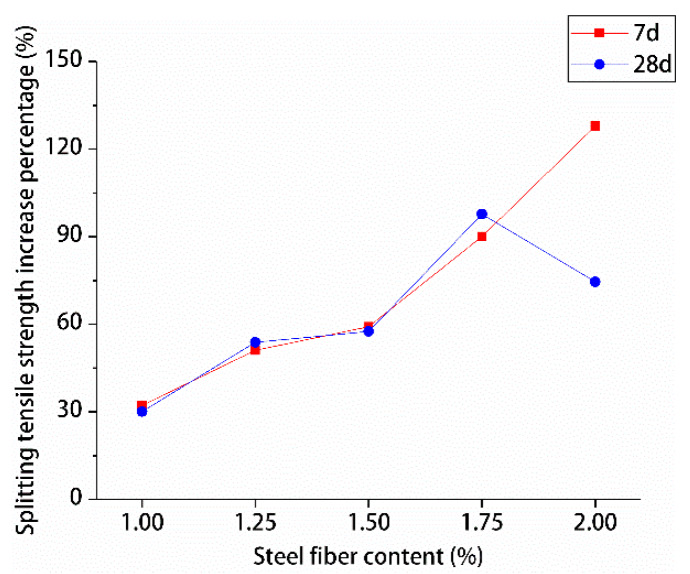
Increase percentage of splitting tensile strength.

**Figure 17 materials-13-02243-f017:**
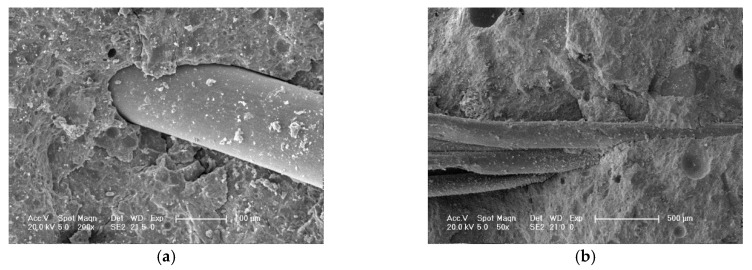
Microscopic picture of hybrid polypropylene-steel fiber RPC: (**a**) Interface transition zone between steel fiber and matrix and (**b**) steel fiber knot.

**Figure 18 materials-13-02243-f018:**
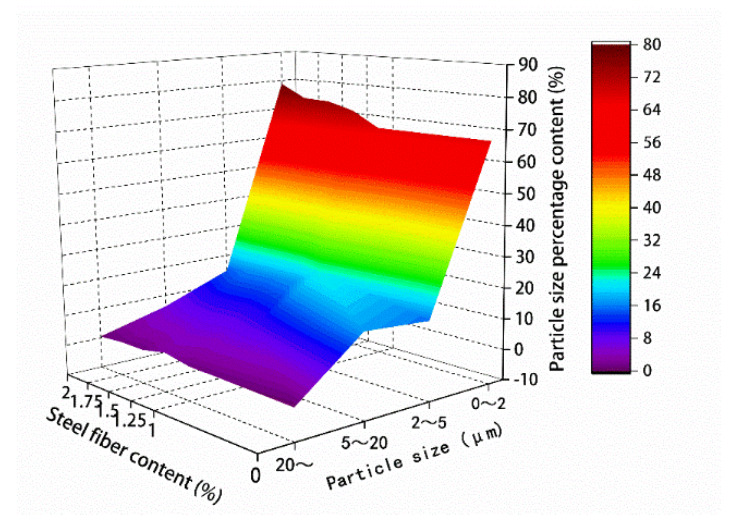
Particle size percentage content.

**Figure 19 materials-13-02243-f019:**
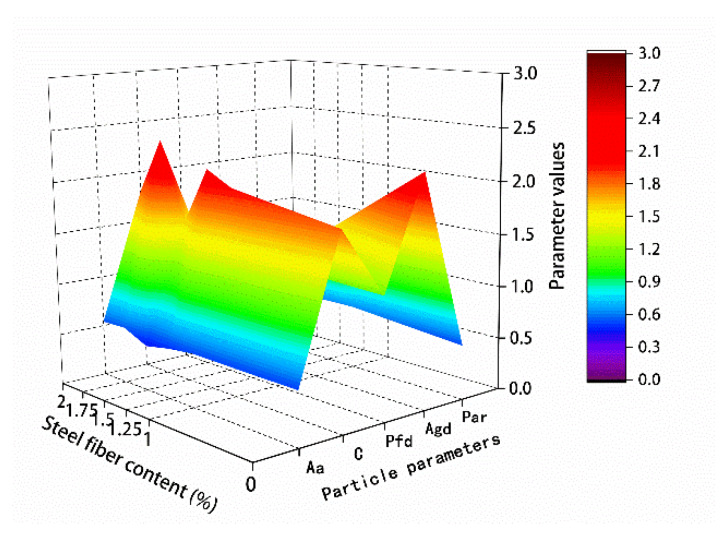
Microparticle parameter.

**Figure 20 materials-13-02243-f020:**
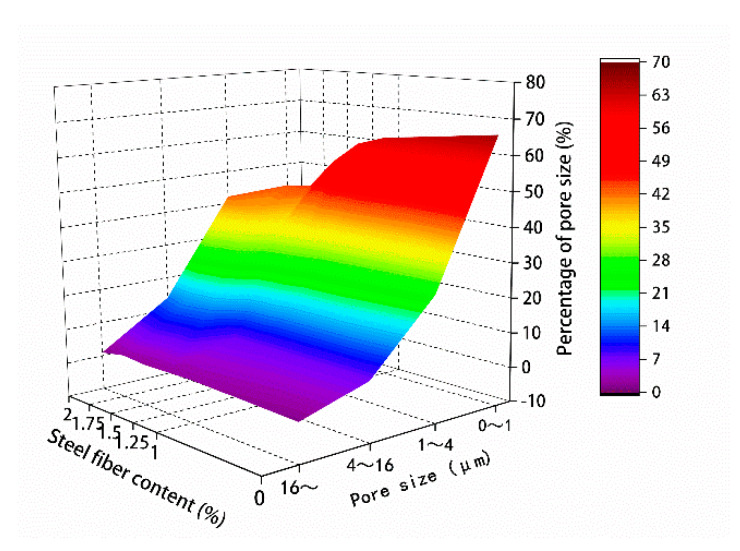
Percentage of pore size.

**Figure 21 materials-13-02243-f021:**
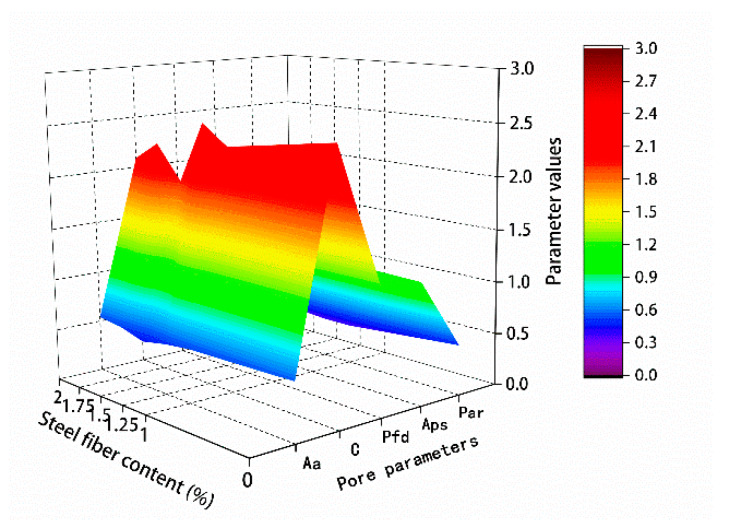
Pore microparameter.

**Figure 22 materials-13-02243-f022:**
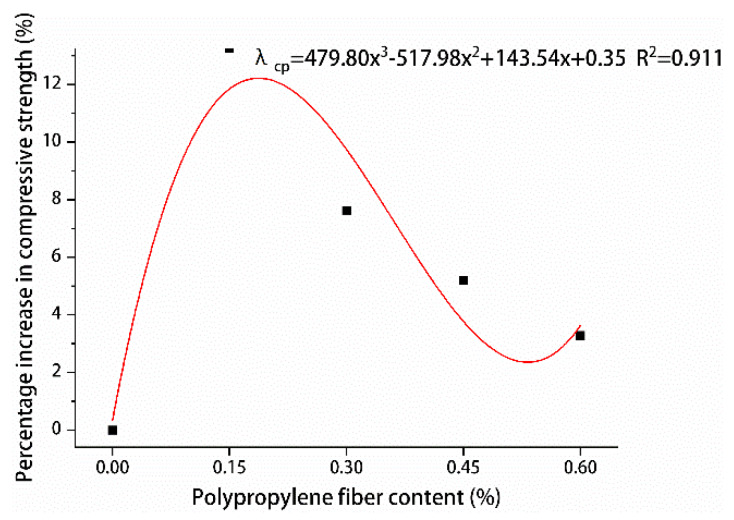
Relationship between compressive strength and polypropylene fiber content.

**Figure 23 materials-13-02243-f023:**
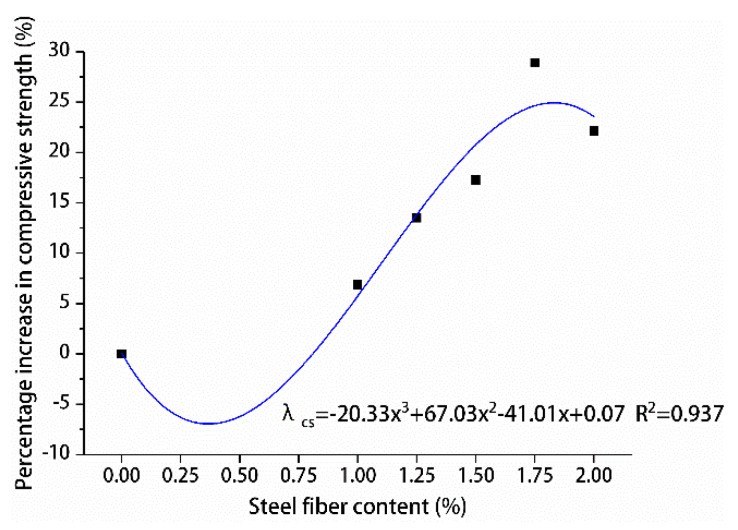
Relationship between compressive strength and steel fiber content.

**Figure 24 materials-13-02243-f024:**
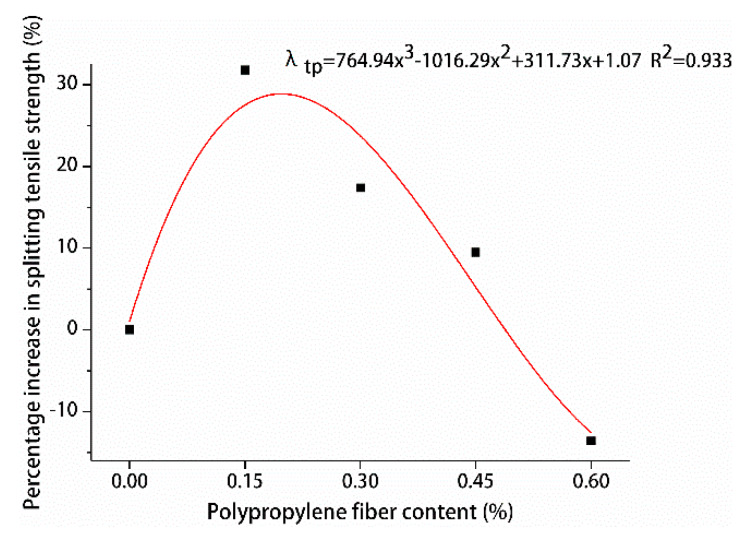
Relationship between splitting tensile strength and polypropylene fiber content.

**Figure 25 materials-13-02243-f025:**
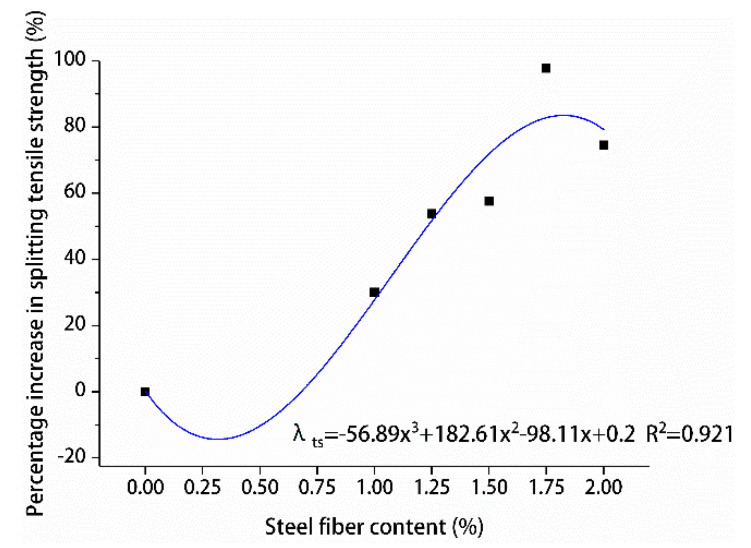
Relationship between splitting tensile strength and steel fiber content.

**Table 1 materials-13-02243-t001:** Physical and chemical properties of cement.

Properties		Standard Value	Actual Value
Physical properties	Specific surface area (m^2^/kg)	≥300	367
	Initial set (min)	≥45	99
	Final set (min)	≤390	145
Compressive strength	3 day (MPa)	≥23.0	29.1
	7 day (MPa)	≥52.5	
Flexural strength	3 day (MPa)	≥4.0	6.0
	7 day (MPa)	≥7.0	
Chemical properties	Stability	Qualified	Qualified
	Loss on ignition (%)	≤3.5	1.61
	MgO (%)	≤5.0	0.98
	SO_3_ (%)	≤3.5	2.62
	Insolubles (%)	≤1.5	1.01
	Cl^−^ (%)	≤0.06	0.007

**Table 2 materials-13-02243-t002:** Physical and chemical properties of silica fume.

Properties		Standard Value	Actual Value
Physical properties	Specific surface area (m^2^/kg)	≥15	20
	Pozzolanic activity index (%)	≥85	116
Chemical properties	SiO_2_ (%)	≥85.0	94.5
	Loss on ignition (%)	≤6.0	2.5
	Cl^−^ (%)	≤0.02	0.02
	Moisture content (%)	≤3.0	1.2
	Water demand ratio (%)	≤125	118

**Table 3 materials-13-02243-t003:** Chemical properties of fly ash.

Properties		Standard Value	Actual Value
Chemical properties	Water demand ratio (%)	≤105	94
	Loss on ignition (%)	≤8.0	1.0
	Moisture content (%)	≤1.0	0.1
	SO_3_ (%)	≤3.0	0.3
	CaO_3_ (%)	≤1.0	0.16
	MgO (%)	≤5.0	1.08
	Cl^−^ (%)	≤0.02	0.01

**Table 4 materials-13-02243-t004:** Physical and chemical properties of superplasticizer.

Properties		Standard Value	Actual Value
Physical properties	Water reduction rate	≥25	28.5
	Gas content (%)	≤6.0	3.2
	Bleeding rate (%)	≤60	50
Chemical properties	Cl^−^ (%)	≤0.1	0.05
	OH^−^ (%)	≤3	1.2
	Na_2_SO_4_	≤0.5	0.06

**Table 5 materials-13-02243-t005:** River sand screening results.

Properties	Sieve Size (mm)				
	1.18	0.6	0.3	0.15	Sieve Bottom
Sieve residue (g)	1	185	215	90	9
Submeter (%)	0.2	37	43	18	1.8
Cumulative (%)	0.2	37.2	80.2	98.2	100

**Table 6 materials-13-02243-t006:** Physical properties of polypropylene fiber and steel fiber.

Fiber Type	Diameter (mm)	Length (mm)	Tensile Strength (MPa)
Polypropylene fiber	0.034	18	358
Steel fiber	0.2	20	2850

**Table 7 materials-13-02243-t007:** Polypropylene fiber reactive powder concrete (RPC) mix proportions.

No.	Polypropylene Fiber Content (%)	Water/Binder Ratio	Cement (kg/m^3^)	Fly Ash (kg/m^3^)	Silica Fume (kg/m^3^)	Sand/Binder Ratio	Number of Specimens
P0	0	0.18	980	280	140	0.75	12
P1	0.15						12
P2	0.3						12
P3	0.45						12
P4	0.6						12

**Table 8 materials-13-02243-t008:** Results of mechanical property test on polypropylene fiber RPC.

No.	Compressive Strength (MPa)		Coefficient of Variation (%)		Splitting Tensile Strength (MPa)		Coefficient of Variation (%)	
	7-Day	28-Day	7-Day	28-Day	7-Day	28-Day	7-Day	28-Day
P0	78.26	84.86	8.46	4.61	6.5	9.78	7.12	4.07
P1	85.12	96.12	3.48	5.21	8.52	12.89	6.23	6.86
P2	83.64	91.33	3.07	5.40	10.02	11.48	4.62	6.92
P3	81.34	89.27	2.10	5.78	8.29	10.71	6.57	3.32
P4	79.03	87.65	4.96	3.03	6.73	8.45	1.35	6.40

**Table 9 materials-13-02243-t009:** Hybrid polypropylene-steel fiber RPC mix proportions.

No.	Polypropylene Fiber Content (%)	Steel Fiber Content (%)	Number of Specimens
SP0	0.15	0	12
SP1		1	12
SP2		1.25	12
SP3		1.5	12
SP4		1.75	12
SP5		2	12

**Table 10 materials-13-02243-t010:** Results of mechanical property test on hybrid polypropylene-steel fiber RPC.

No.	Compressive Strength (MPa)		Coefficient of Variation (%)		Splitting Tensile Strength (MPa)		Coefficient of Variation (%)	
	7-Day	28-Day	7-Day	28-Day	7-Day	28-Day	7-Day	28-Day
SP0	85.12	96.12	4.89	2.54	8.52	12.89	6.48	2.10
SP1	88.39	102.72	1.68	7.59	11.26	16.77	3.54	15.15
SP2	89.01	109.08	4.21	0.66	12.87	19.83	11.65	4.70
SP3	95.26	112.71	4.68	1.85	13.56	20.31	9.52	2.76
SP4	100.96	123.92	2.03	2.35	16.19	25.49	3.31	5.34
SP5	94.85	117.39	5.22	3.49	19.42	22.5	8.32	5.43

**Table 11 materials-13-02243-t011:** Particle size percentage content.

No.	Particle Size Percentage Content (%)			
	0–2 μm	2–5 μm	5–20 μm	>20 μm
SP0	67.18	15.42	17.18	0.22
SP1	67.05	22.53	9.26	1.16
SP2	71.93	17.14	8.13	2.8
SP3	74.22	16.25	8.42	1.11
SP4	74.51	16.52	7.82	1.15
SP5	78.82	14.55	6.18	0.45

**Table 12 materials-13-02243-t012:** Microparticle parameters.

No.	Particle Area Ratio (%)	Average Particle Size (μm)	Granular Fractal Dimension (−)	Circularity (−)	Average Abundance (−)
SP0	0.52	2.20	1.16	1.82	0.51
SP1	0.62	1.39	1.21	2.00	0.52
SP2	0.62	1.29	1.20	2.14	0.49
SP3	0.60	1.17	1.18	1.59	0.44
SP4	0.71	1.14	1.17	2.35	0.55
SP5	0.38	0.93	1.15	1.80	0.54

**Table 13 materials-13-02243-t013:** Percentage of pore size.

No.	Percentage of Pore Size (%)			
	0–1 μm	1–4 μm	4–16 μm	>16 μm
SP0	66.21	25.82	6.87	1.1
SP1	61.92	28.76	7.12	2.19
SP2	59.29	31.16	7.53	2.01
SP3	52.88	32.93	14.38	1.79
SP4	44.09	39.09	15.45	2.36
SP5	43.14	42.15	13.73	0.98

**Table 14 materials-13-02243-t014:** Pore microparameters.

No.	Pore Area Ratio (%)	Average Pore Size (μm)	Pore Fractal Dimension (−)	Circularity (−)	Average Abundance (−)
SP0	0.48	1.16	1.20	2.51	0.55
SP1	0.38	1.19	1.21	2.34	0.52
SP2	0.38	1.27	1.2	2.53	0.49
SP3	0.40	1.31	1.17	1.94	0.44
SP4	0.29	1.33	1.18	2.27	0.51
SP5	0.62	1.51	1.2	2.09	0.54

**Table 15 materials-13-02243-t015:** Compressive strength of hybrid polypropylene-steel fiber RPC at 28 days.

Polypropylene Fiber Content (%)	$\lambda_{cp}\;(\%)$	Steel Fiber Content (%)	$\lambda_{cs}\;(\%)$	Calculated Value (MPa)	Actual Value (MPa)	Error (%)
0.15	1.12	0	1.00	94.98	96.12	−1.20
		1	1.06	100.38	102.72	−2.33
		1.25	1.14	108.04	109.08	−0.96
		1.5	1.21	114.61	112.71	1.66
		1.75	1.25	118.28	123.92	−4.76
		2	1.23	117.24	117.39	−0.12

**Table 16 materials-13-02243-t016:** Hybrid polypropylene-steel fiber RPC compressive strength [35].

Polypropylene Fiber Content (%)	$\lambda_{cp}\;(\%)$	Steel Fiber Content (%)	$\lambda_{cs}\;(\%)$	Calculated Value (MPa)	Actual Value (MPa)	Error (%)
0.1	1.10	0.5	0.94	98.21	103.41	−5.29
		1	1.06	110.76	113.56	−2.53
		1.5	1.21	126.46	125.47	0.79
		2	1.23	129.36	138.13	−6.78
0.15	1.12	0.5	0.94	99.86	107.36	−7.52
		1	1.06	112.61	109.30	2.94
		1.5	1.21	128.58	143.54	−11.63
		2	1.23	131.53	140.27	−6.64
0.2	1.12	0.5	0.94	100.15	101.95	−1.80
		1	1.06	112.94	103.32	8.52
		1.5	1.21	128.96	144.68	−12.19
		2	1.23	131.92	128.77	2.39
0.25	1.11	0.5	0.94	99.42	105.51	−6.13
		1	1.06	112.12	107.72	3.92
		1.5	1.21	128.02	145.59	−13.72
		2	1.23	130.96	148.51	−13.40

**Table 17 materials-13-02243-t017:** Splitting tensile strength of hybrid polypropylene-steel fiber RPC at 28 days.

Polypropylene Fiber Content (%)	$\lambda_{tp}\;(\%)$	Steel Fiber Content (%)	$\lambda_{ts}\;(\%)$	Calculated Value (MPa)	Actual Value (MPa)	Error (%)
0.15	1.28	0	1.00	12.50	12.89	−3.13
		1	1.28	15.94	16.77	−5.19
		1.25	1.52	18.93	19.83	−4.74
		1.5	1.72	21.44	20.31	5.28
		1.75	1.83	22.81	25.49	−11.75
		2	1.80	22.37	22.5	−0.60

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
