# Peer review of "Study on Mechanical Properties of Hybrid Polypropylene-Steel Fiber RPC and Computational Method of Fiber Content"

_materials, 2020, doi:10.3390/ma13102243_

Round 1

Reviewer 1 Report

The article covers the topic of the Relationship Between the Mechanical Properties and the Microscopic Parameters of Polypropylene–Steel
Fiber RPC Based on Grey Relational Theory. The subject and the supporting experiments present added value to the body of knowledge on the subject area. The topic of the article is in scope of journal. However, in my opinion article should be thoroughly improved before publication.
The following modification should be considered:

1. In my opinion, the major inacurracy in this article is involved with definition of reactive powder concrete (RPC). In general, when we considering RPC, the first association is involved with the specific materails used to production, technology, ultra compressive/tensile strength and special properties of this concrete. In this work authors produced and examined concrete where the compressive strength is c.a. 90 MPa only. Please clearly clarify me and in the manuscript the reason, why you called concrete which you have made as "reactive powder concrete"?
2. I suggest to add point 2 - Research significance - Please descibe here the main essence of the experiment (especially text in lines 87-95).
3. I sugget that literature survey should contain more thorough content. The shape of the coarse aggregate grains is very important aspect influencing the strength of concrete with and without fibres. Consider to use some literature involved with this scientific area where authors provided the microparticle parameters,
such as: https://doi.org/10.3390/ma11081372; https://doi.org/10.1016/j.conbuildmat.2019.117794.
4. I think that figure 1 could be more legible if the horizontal axis will be modified (values since 0 to 1.2); according the Cartesian system.
5. What type of plasticizer was used? On what phenomenon the action was based? Steric effect/dispersion or another?
6. Why the time of mixing with plasticizer was only 2 minutes? It should be 5 minute at least - admixtures needs time to  show its action.
7. In figure 3 I suppose that phrase 'no' should be connected between 'whether or not steel fiber' and 'water + superplasticizer'.
8. Table 7 - please show the content by mass - please change the content for the unit kg/m3 and add columns: number of specimens and fibre content.
9. Please show all tested specimens.
10. In table 9 please show standard deviations of the strength.
11. In table 13 please add the units. If some parameter does not have unit please show '[-]'.
12. Did you measure axial/transverse displacements? It would be indicated to show the axial and transverse strains.

Author Response

Thank you for your advice. All your suggestions are very important and have important guiding significance for my thesis writing and scientific research work. I have revised the manuscript on the basis of your suggestions and carefully proofed the manuscript. I modified the manuscript according to your suggestion and uploaded the modified content to the attachment.

Reviewer 2 Report

This research paper explains about relationship between the mechanical properties and the microscopic parameters of polypropylene–steel fiber RPC based on grey relational theory. -Abstract: The text must be carefully revised. Some sentences contain mistakes (in the abstract: very general statements) whereas some sentences must be reworded as the English is “meaningless”. I strongly recommend that the authors retain the services of a professional editor. There are many reputable companies that offer these services. - Introduction is poorly written. Proper references need to be used rather than using others. Language can be improved. The sentences are half constructed or incomplete in a way that the readers are expected to fend for themselves in order to understand their meaning. Line 62: Splitting tensile strength is not a form of compressive strength. Line 74: With increase in what temperature? Lines 75 – 77 do not make sense. Line 98: The cement brand does not have to be mentioned. Likewise the companies from which silica fume, super-plasticizer, fibres and fly ash are procured need not be mentioned. Figure 3. The flowchart does not make any sense Figure 4 and 5 are not representative of the failure pattern of the two types of concrete. They seem to be presented in a way to convey a point which seems to be different than which was observed. Line 201: The reasons given for the fall in performance are not connected. They are independent. The tensile failure pattern has to be revisited. Literature can be read before drawing conclusions which are not logical. Overall the paper appears to be of satisfactory quality. With these minor corrections the paper can be accepted for publication.

Author Response

(The authors gave the same response as above.)

Reviewer 3 Report

The authors reported the study on Relationship Between the Mechanical Properties and the Microscopic Parameters of  RPC. The main conclusions presented in the paper are well supported by the figures and supporting text, and therefore it is recommended that the manuscript can be published in Materials. However, to meet the journal quality standards, the following major comments need to be addressed

Specific major comments and suggestions:

1. Introduction writing part can be improved. The novelty should be highlighted in introduction. Authors should add more references for comparison and clearly highlight the advancement of their research with existing published results.

2. Authors are encouraged to review similar article in different length scale related to the model to enrich the content for broad audience. For example, Phase-field method is very useful methods to predict macroscopic mechanical properties and microstructure evolution of mutiphase material such as  concrete: Physical Review B, 2015, Vol. 91, 174109 ; DOI: 10.1103/physrevb.91.174109 , Acta Materialia 2016, Vol. 109 DOI: 10.1016/j.actamat.2015.12.013; regularized cohesive zone model (CZM) and size effect of concrete and failure : Engineering Fracture Mechanics197, 66-79, Engineering Fracture Mechanics 208 (2019): 151-170. Authors should discuss  them in the introduction section.

3. There are some grammatical mistakes and drawbacks in the manuscript, Please double check it.

4. Authors may add "computational methods" chapter the way so anybody can repeat your computational procedures, like a recipe.

5. The authors may add a new table to compare their results with published articles.(optional)

6. Results and discussion: - Comparison with existing literature can be improved, limitations of the results should be discussed, future research should be outlined.

7. Conclusions: clearly state whether your hypotheses were confirmed or not.

Author Response

(The authors gave the same response as above.)

Reviewer 4 Report

The topic of the manuscript is interesting, and such research is of great importance to the research community.

After careful reading of the manuscript, I can conclude that there are many shortcomings that should be corrected before publishing this research in a journal.

Previous relates primarily to the technical arrangement and spelling. For example, on page three in line 102 and line 113, the author refers to the pictures and tables in the wrong way. I suggest the authors carefully read the manuscript and remove all technical and spelling errors.

The research presented in this manuscript is designed as an experimental study with the idea of contributing to the development of economic and reasonable reactive powder concrete.

In the first part of this study, different dosages of polypropylene fibres RPC of varying compressive strengths and splitting tensile strengths were studied to determine the optimum content of polypropylene fibres RPC.

In second part of this study, the optimum dosage of polypropylene fibres was combined by adding and varying contents of steel fibres. The steel fibres content of polypropylene was studied in terms of compressive strength and splitting tensile strength.

In third part of this study, the mechanical properties and micro parameters of polypropylene steel fibres RPC were evaluated on the basis of grey relation theory.

The third part of the paper, Section 3.4, which deals with correlation analysis, should, in my opinion, be excluded from this manuscript and therefore change the title of this manuscript. This part unnecessarily burdens the manuscript and brings nothing new. Due to the length of the work, I think that with some improvements and extensions, this section might bring something new, but then it should be some new manuscript.

With some clarifications and corrections, the remaining part the manuscript may be published in a journal.

Regarding the experimental part of the paper, the following are some concerns that should be eliminated before publishing in the journal.

I think it is missed to mention in the introduction that the characteristics of micro-reinforced concrete also depend significantly on the rate of deformation and that the fibres significantly increase the ductility. So the authors in works such as: (http://www.scielo.br/scielo.php?pid=S1679-78252018000200500&script=sci_arttext) have shown that the fibres have significantly increased ductility of the specimens under static compressive failure, as well as both its splitting and flexural tensile strength.

The Sample Preparation section should indicate how the samples are processed before testing. It is necessary to describe to what level the surface that is in contact with the press is polished. According to most authors, differences in this treatment can lead to significant dissipation of results for the same concrete mix, especially on small samples.

The average of three measurements was used as the compressive strength and splitting tensile strength of the test specimens. Since these are three samples, it would be good to indicate the standard deviation for each sample set (for each strength).

Of course, for micro-reinforced concrete, the way of mixing and the way of preparation of samples significantly affect the arrangement of fibres within the concrete sample and therefore the mechanical characteristics. I wonder if you somehow considered this. Did you repeat some of the experiments (testing three samples for some strength) and did you get the same results?

The authors state that the goal is to determine the optimum content of polypropylene fibres in RPC on the basis of the study of different steel fibres contents and the mechanical behaviour of steel fibres and polypropylene RPC. The manuscript shows only the compressive strength and the tensile strength, and there is no information on mechanical characteristics that would also be of interest (Modulus of Elasticity, deformability, Poisson coefficient, etc.). It is not clear why?

In Chapter 3.2. Experimental results and analysis of the mechanical properties of polypropylene – steel fibres RPC authors state: On the basis of the optimal content of polypropylene fibres RPC, 0%, 1%, 1.25%, 1.5%, 1.75% and 2% of steel fibres were added. The mechanical properties of polypropylene–steel fibres RPC were further studied.

Did you combined here steel fibres only with content of polypropylene fibres of 0:15% in RPC Is it possible that some other amount of polypropylene fibres works better in combination with steel fibres although this amount alone (concrete mix without steel fibres) gives the highest strength?

Author Response

(The authors gave the same response as above.)

Author Response

(The authors gave the same response as above.)

Round 2

Reviewer 1 Report

All remarks have been considered by authors. Errors have been eliminated. The authors responded to all comments of the reviewer.
The current version is satisfactory for reviewer.
In my opinion, article could be published.

Author Response

Thanks reviewer for good comments and hard work.

Reviewer 3 Report

Authors did address most of the reviewer previous comments. However, some of the comments were not satisfactorily taken care off. Hence, this revised manuscript  still needs revision. It is not acceptable in the current form. Hence following major points needed to be addressed before publication.

Major comments :

Reviewer previous comment [1]: Authors were asked to highlight the novelty of the work and the advancement of their research with
existing published results. This is still  not taken care properly in  the current revision. They must address this in order to make this manuscript scientifically impactfull.

Reviewer previous comment [2]: Authors were asked to add some specific references and compare them. In reply, authors mentioned "a review has been made according to your suggestions" , but it not incuded in the main text. Authors should add a future research scope of the current work  and add those references  in different length scale related to the model to enrich the content for broad audience.

Author Response

Thank you for your advice. All your suggestions are very important and have important guiding significance for my thesis writing and scientific research work. I have revised the manuscript on the basis of your suggestions and carefully proofed the manuscript. The following is a description of the changes based on your comments:

Response 1: According to your suggestion, it has been modified as follows:

Reactive powder concrete (RPC) is a new type of building material with high strength and toughness developed by Richard in the 1990s. It removes coarse aggregate on the basis of ordinary concrete, and optimizes the compactness of aggregate with quartz sand, improving the homogeneity of matrix, and reducing porosity by adding active mineral admixtures such as silica fume and fly ash, while improving the microstructure by thermal curing [1–3]. In order to achieve ultrahigh strength of RPC, current research uses quartz sand [4–6] to prepare RPC through heat curing [7] or autoclave curing [8–9]. Thermal and autoclave curing can effectively improve RPC mechanical properties and promote its hydration process of RPC and enhance pozzolanic activity [10–11]. However, the cost of preparing RPC is increased because quartz sand cost is more expensive. The construction appears to be more difficult because of field thermal curing.These problems have limited the application of RPC and in order to expand its application , low cost common materials and conventional curing methods are adopted at the expense of certain reduction of strength.

Polypropylene fiber is a kind of non-toxic harmless safe material, which is cheap and have relatively low mass. Adding polypropylene fiber into RPC not only could improve strength and toughness, but also its crack resistance [12,13]. A lot of researches suggests that polypropylene fiber RPC has limited effect on the compressive strength [14–15], while significantly improving the ductility of the RPC and preventing explosive spalling [16–17]. When the content from 0.3% to 0.9%, the addition of polypropylene fiber can improve the mechanical properties of RPC. When the dosage of more than 0.9%, the mechanical properties of RPC decreased to a certain extent [18]. Incorporation of steel fibers can significantly improve the mechanical properties of the RPC and change its failure morphology, and the ductility is significantly improved [19]. It is widely used in the preparation of RPC, the addition of steel fiber can reduce cracks and inhibit its broadening. However, the content of steel fiber should be paid attention. If the amount of steel fiber is too high, it will cause the steel fiber to occupy part of the mixing water, and the steel fiber lacks enough slurry to wrap and fill, making the steel fiber unevenly distributed and agglomerated, which in turn leads to the steel fiber's contribution to improving strength did not increase, but decrease [20]. Due to the different curing methods of RPC and the different sizes of steel fibers, the obtained optimal amount of steel fiber are also different. when the amount of steel fiber is increased from 0% to 3%, the compressive strength of RPC is slightly increased [21]. However, after the amount of steel fiber is more than 3.5%, the compressive strength of RPC does not increase any more. The splitting tensile strength increases obviously when the steel fiber is less than 2%, and the splitting tensile strength remains almost unchanged over 2% [22]. The results showed that the splitting tensile strength increased from 4.033 MPa to 9.609 MPa, 11.944MPa and 12.896 MPa, respectively when the steel fiber increased from 0% to 1%, 1.5% and 2% [23]. The results showed that when the content of steel fiber from 0% to 4%, the compressive strength and splitting tensile strength are increased, and the compressive strength of steel fiber to RPC is increased less, and the splitting tensile strength is obviously improved [20,24]. Moreover, the effect of incorporation of different types of steel fibers on the mechanical properties of the RPC is also different. When 3% straight steel fibers is used, the compressive strength can greater than 150 MPa, while the compressive strength of 3% end hook and corrugated fiber RPC 28d is 48% and 59% higher than that of straight steel fibers, respectively [25]. Hence, it is necessary to further study the mechanical properties of polypropylene-steel fiber RPC. 

Concrete is a complex multi-phase heterogeneous material, and factors such as temperature and stress will induce phase transformation [26–27]. It is difficult to make reliable analysis of concrete only through macroscopic mechanical properties. Therefore, it is necessary to analyze the macroscopic properties of concrete through microscopic research to determine the relationship between the microstructure and macroscopic properties of concrete, which is of great significance to the research and development of new concrete materials. Studies have shown that the particle morphology and pore parameters of aggregate have a significant effect on the compressive strength of concrete [28–29]. Some scholars use XCT combined phase field theory to analyze the microscopic parameters of materials, but the computational cost involved is high [30]. The IPP (Image–pro plus) software has powerful image processing function, which is not only simple to operate but also low cost. Therefore, the microscopic parameters of polypropylene-steel fiber RPC are analyzed by combining scanning electron microscope (SEM) with image-pro plus (IPP) software (SEM–IPP).

In summary, previous studies mainly focus on the RPC mechanical properties of single-doped polypropylene fiber and single–doped steel fiber, and the mechanical properties of hybrid polypropylene–steel fiber RPC are less studied. Since the hybrid fibers have certain complementarity and synergy, the high–modulus steel fibers can effectively enhance the mechanical properties of RPC, while the low–modulus polypropylene fibers have certain advantages in improving the toughness of RPC. On the basis of determining the optimum content of polypropylene fiber RPC, the influence of different steel fiber content on the mechanical properties of hybrid polypropylene–steel fiber was studied. By combining SEM with IPP software, the particle and pore parameters of hybrid polypropylene-steel fiber RPC were analyzed and computing method of content of polypropylene fiber and steel fiber, as well as the 28 days compressive strength and splitting tensile strength of RPC. According to different engineering requirements, the content of polypropylene fiber and steel fiber content can be selected flexibly, which can provide reference for the preparation of economical and reasonable RPC in practical engineering.

Response 2:According to your suggestion, it has been modified as follows:

Concrete is a complex multi-phase heterogeneous material, and factors such as temperature and stress will induce phase transformation [26-27]. It is difficult to make reliable analysis of concrete only through macroscopic mechanical properties. Therefore, it is necessary to analyze the macroscopic properties of concrete through microscopic research to determine the relationship between the microstructure and macroscopic properties of concrete, which is of great significance to the research and development of new concrete materials. 

In order to provide a reliable reference for the structural design of hybrid polypropylene-steel fiber RPC, in future work, the hybrid polypropylene-steel fiber RPC elastic modulus, deformability and poisson coefficient should be studied. The size effect has an important influence on the destruction process of concrete components [36]. It is necessary to further study the effect of the size effect of hybrid polypropylene-steel fiber RPC components, which has important practical significance for the structural design of hybrid polypropylene-steel fiber RPC.

Thanks reviewer for good comments and hard work.

Reviewer 4 Report

I think the manuscript is improved enough that it can be published in a journal.

Author Response

(The authors gave the same response as above.)

Reviewer 5 Report

Comared to the previous version, the authors improved the manuscript that deserves to be published now. However, as stated in the first revision, the authors must consider that the methologies adopted to assess the mechanical behavior of fiber reinforced RPC, especially in tension, are not totally appropriate. In fact, one of the key points of the paper concerns the assessment of the ability of fibers to improve the tensile toughness and ductility of RPC. Splitting tests performed under load control as well as the  observation of damage evolution in compression are not enough to assess the toughness improvement. The authors must take these considerations into account when they will design future studies.  

Author Response

Your Suggestions will be of great help to my future research.

Thanks reviewer for good comments and hard work.